# Biology and Management of High-Grade Chondrosarcoma: An Update on Targets and Treatment Options

**DOI:** 10.3390/ijms24021361

**Published:** 2023-01-10

**Authors:** Camille Tlemsani, Frédérique Larousserie, Sixtine De Percin, Virginie Audard, Djihad Hadjadj, Jeanne Chen, David Biau, Philippe Anract, Benoit Terris, François Goldwasser, Eric Pasmant, Pascaline Boudou-Rouquette

**Affiliations:** 1Department of Medical Oncology, Cochin Hospital, Paris Cancer Institute CARPEM, Université Paris Cité, APHP.Centre, 75014 Paris, France; 2INSERM U1016-CNRS UMR8104, Cochin Institute, Paris Cancer Institute CARPEM, Université Paris Cité, APHP.Centre, 75014 Paris, France; 3Department of Pathology, Cochin Hospital, Paris Cancer Institute CARPEM, Université Paris Cité, APHP.Centre, 75014 Paris, France; 4Department of Orthopedic Surgery, Cochin Hospital, Paris Cancer Institute CARPEM, Université Paris Cité, APHP.Centre, 75014 Paris, France; 5Department of Genetics, Cochin Hospital, Paris Cancer Institute CARPEM, Université Paris Cité, APHP.Centre, 75014 Paris, France

**Keywords:** (MeSH terms): chondrosarcoma, dedifferentiated, mesenchymal, IDH1/2, novel agents, treatment

## Abstract

This review provides an overview of histopathology, clinical presentation, molecular pathways, and potential new systemic treatments of high-grade chondrosarcomas (CS), including grade 2–3 conventional, dedifferentiated, and mesenchymal CS. The diagnosis of CS combines radiological and histological data in conjunction with patient clinical presentations. Conventional CS is the most frequent subtype of CS (85%) and represents about 25% of primary bone tumors in adults; they can be categorized according to their bone location into central, peripheral, and periosteal chondrosarcomas. Central and peripheral CS differ at the molecular level with either *IDH1/2* mutations or *EXT1/2* mutations, respectively. *CDKN2A/B* deletions are also frequent in conventional CS, as well as *COL2A1* mutations. Dedifferentiated CS develops when low-grade conventional CS transforms into a high-grade sarcoma and most frequently exhibits features of osteosarcoma, fibrosarcoma, or undifferentiated pleomorphic sarcoma. Their molecular characteristics are similar to conventional CS. Mesenchymal CS is a totally different pathological entity exhibiting recurrent translocations. Their clinical presentation and management are different too. The standard treatment of CSs is wide en-bloc resection. CS are relatively radiotherapy resistant; therefore, doses >60 Gy are needed in an attempt to achieve local control in unresectable tumors. Chemotherapy is possibly effective in mesenchymal chondrosarcoma and is of uncertain value in dedifferentiated chondrosarcoma. Due to resistance to standard anticancer agents, the prognosis is poor in patients with metastatic or unresectable chondrosarcomas. Recently, the refined characterization of the molecular profile, as well as the development of new treatments, allow new therapeutic options for these rare tumors. The efficiency of IDH1 inhibitors in other malignancies suggests that these inhibitors will be part of *IDH1/2* mutated conventional CS management soon. Other treatment approaches, such as PIK3-AKT-mTOR inhibitors, cell cycle inhibitors, and epigenetic or immune modulators based on improving our understanding of CS molecular biology, are emerging.

## 1. Introduction

Chondrosarcomas (CS) are rare mesenchymal neoplasms that are defined by the production of an abnormal cartilaginous matrix. They define a diverse group of sarcomas with varied morphological features and different clinical behaviors. CSs account for 25% of primary bone tumors, with about 150 new diagnosed cases per year in France [1]. These are mostly adult tumors.

The most common locations of CS include the pelvis and the proximal femur. CS can be endo-medullary or grow on the bone surface. They can develop in a healthy bone or occur in a pre-existing bone lesion (chondroma or osteochondroma, rarely in Paget’s disease). Some radiation-induced CS have been described in old series or in rare case reports. They most likely represent chondroblastic osteosarcomas [2,3]. In the context of enchondromatosis (Ollier’s disease and Maffucci syndrome), the risk of malignant transformation of enchondromas is increased (46% and 57%, respectively) [4].

The diagnosis of CS is based on a multidisciplinary approach considering radio-pathological correlations in conjunction with patient clinical presentation. Standard radiograph and computed tomography (CT) often suggest the diagnosis of CS because of the identification of typical “ring-and-arc” chondroid matrix mineralization and aggressive features (deep endosteal scalloping and soft-tissue extensions). Additional imaging modality, including magnetic resonance imaging (MRII), is frequently employed to evaluate staging and guide surgical resection.

Conventional CS is the most common pathological subtype. Surgical excision is the primary treatment modality of CS. The treatment of conventional CS is problematic in unresectable or metastatic diseases because of its primary resistance to standard sarcoma chemotherapy regimens. 

Other subtypes of CS are significantly less common. Dedifferentiated CS are biphasic tumors with a low-grade conventional CS component juxtaposed to a non-cartilaginous high-grade sarcoma, most frequently exhibiting features of osteosarcoma, fibrosarcoma, or undifferentiated pleomorphic sarcoma. Mesenchymal CS (MCS) is a different pathological entity. It is a highly malignant tumor exhibiting a bimorphic histological pattern, with an undifferentiated small round cell component admixed with islands of well-differentiated cartilage. Chemotherapy is possibly effective in mesenchymal chondrosarcoma and is of uncertain value in grade 3 conventional chondrosarcoma and in dedifferentiated chondrosarcoma; these subtypes are rare and bear a poor prognosis.

Understanding the different histological subtypes of CS is helpful in predicting biological behavior. In this review, a summary is given about CS clinical behavior, pathological characteristics, and current treatment modalities. In the following, we will also discuss the molecular pathways in high-grade CS, including conventional CS, dedifferentiated CS, and mesenchymal CS, and a schematic illustration of key signaling pathways underpinning CS genesis. Furthermore, published and ongoing clinical trials for CS patients will also be presented.

## 2. Clinical, Pathological and Molecular Characteristics

There are several CS subtypes (Table 1). We will describe all the subtypes according to the 2020 WHO classification of tumors on bone and soft tissues, which notably identifies conventional chondrosarcoma, dedifferentiated chondrosarcoma, and mesenchymal chondrosarcoma. We will highlight the main clinical and molecular characteristics of these subtypes. We will not describe in this review clear cell chondrosarcoma, which is commonly considered a low-grade tumor, is less aggressive and usually treated with surgery alone, as well as extraskeletal myxoid chondrosarcoma, which is a soft tissue sarcoma (Table 1).

### 2.1. Conventional Chondrosarcoma

Conventional chondrosarcoma is the most frequent subtype of chondrosarcoma (85%) and represents about 25% of primary bone tumors. Conventional CSs can be categorized according to their bone location in central and peripheral CSs [5] (Figure 1). The vast majority (±85%) are central CSs, designated as such based on their central location within the medullar cavity. Peripheral chondrosarcomas are intermediate-grade to the high-grade malignant cartilaginous matrix–producing neoplasms that originate from the bone surface of a pre-existing osteochondroma. These tumors most commonly arise in osteochondromas of the pelvis, trunk, and proximal femur. Finally, a minority (<1%) occurs on the surface of the bone, in between the cortical bone and the periosteum, and invades the underlying cortex or is >5 cm. This latter entity is named periosteal CS [6].

Chondrosarcomas can be primary or secondary tumors. Indeed, enchondromas and osteochondromas can evolve towards central or peripheral CSs, respectively (Figure 1). Central and peripheral CSs differ at the molecular genetic level. *Isocitrate dehydrogenase 1/2* (*IDH1* or *IDH2*) mutations are present in 85% of hereditary enchondromatosis-associated disorders, Ollier disease (enchondromatosis only), Maffuci syndrome (enchondromatosis and hemangiomas) and 50% of solitary enchondromas, whereas the biallelic inactivation of the exostosin glycosyltransferase (*EXT1* or *EXT2*) genes is observed in the majority of both sporadic and hereditary osteochondromas (Figure 1). These mutations are early events, and similar alterations are found in chondrosarcomas with the presence of *IDH1/2* mutations in central CS and *EXT1/2* mutations in peripheral CS. *IDH1/2* mutations are found in primary or secondary tumors. Interestingly, *EXT1/2* mutations seem to be less frequently detected in sporadic secondary peripheral chondrosarcomas than expected based on the assumption that they originate in sporadic osteochondromas, in which the homozygous inactivation of *EXT1* is found in about 80% of our cases. De Andrea and colleagues suggest that wild-type cells with functional EXT are predisposed to acquire other mutations, giving rise to secondary peripheral chondrosarcoma and indicating that EXT-independent mechanisms are involved in the pathogenesis of secondary peripheral chondrosarcoma [7]. The genetic background of periostal CS is less known. A cohort of 38 samples identified no *EXT1/2* mutations and *IDH1/2* mutations in only 15% of cases suggesting a supposed relationship with central CS [8].

The mutations of *IDH1/2* are present in 30–90% of central CS. In addition to *IDH1/2* mutations, the collagen type II alpha-1 gene (*COL2A1*) mutations are found in about 40% of central CS. *COL2A1* encodes the α-chain of type II collagen fibers: the major collagen constituent of articular cartilage. Interestingly, *COL2A1* mutations are specific to chondrosarcomas. No *COL2A1* mutations have been found in other bone sarcomas. Cyclin-dependent kinase inhibitor 2A/B (*CDKN2A* and *CDKN2B*) deletions are also recurrent in CS chondrosarcomas and occur in 75% of high-grade central CS. They are not present in low-grade cartilaginous tumors. Moreover, the *CDKN2A/B* copy number variation is seen in both *IDH1/2* wild-type and mutant central CS with enrichment in *IDH1/2* mutated tumors [9].

Chondrosarcomas can progress from low grade to high grade, which is reflected by their increased cellularity, nuclear atypia, myxoid changes, and increased vascularization. Central and peripheral CS are histologically subdivided into atypical cartilaginous tumors (ACT), grade 1, grade 2, and grade 3 CS. In the 2020 classification, it was proposed that ACT should be restricted to tumors arising in the appendicular skeleton, which are more amenable to complete surgical resection than tumors located in the axial skeleton. Conversely, « chondrosarcoma, grade 1 » should be used for axial tumors [10]. The reported 5-year overall survival rate is 74% and 31% for grade 2 CS and grade 3 CS, respectively. The 10-year overall survival rate is 62% for grade 2 and 26% for grade 3 tumors [11]. A high grade (2 or 3) was highly related to the increased risk of metastatic disease (29.3% at 5 years vs. 4.6% for Grade 1 tumors; *p* = 0.0001, log-rank test) in a Mayo Clinic descriptive cohort of 344 primary conventional CS patients [12]. Chondrosarcomas in an axial location have significantly lower survival than chondrosarcomas of the extremities. It is unclear whether the *IDH1/2* mutations are associated with the outcome [13,14].

### 2.2. Dedifferentiated Chondrosarcoma

Dedifferentiated chondrosarcomas represent approximately 10% of all reported cases of chondrosarcomas. Dedifferentiated chondrosarcoma is a high-grade subtype of chondrosarcoma with the bimorphic histological appearance of a conventional chondrosarcoma component with an abrupt transition to a high-grade, non-cartilaginous sarcoma (Figure 1). Central dedifferentiated CS and central conventional CS showed similar molecular characteristics and identical tumor suppressor p53 (*TP53*) and *IDH1/2* mutations in their conventional and dedifferentiated components, indicating a common origin. PD-L1 expression has been reported in approximately 50% of dedifferentiated chondrosarcomas [15].

Most patients are older than 50 years at diagnosis. The majority of these tumors are symptomatic at diagnosis, with a pathological fracture in about 20% of patients [16]. The prognosis of patients with dedifferentiated chondrosarcoma is uniformly poor, whatever the treatments used. Most patients die of distant metastases within 2 years of the initial diagnosis [17,18].

### 2.3. Mesenchymal Chondrosarcoma

In 1959, Liechtenstein and Berstein described two « frankly malignant mesenchymal tumors showing focal chondroid differentiation ». They named this tumor mesenchymal chondrosarcoma. Mesenchymal chondrosarcoma is a rare, high-grade, bimorphic sarcoma composed of undifferentiated small round cells and variable amounts of well-differentiated hyaline cartilage. Immunohistochemically, the small round tumor cells stain positive for NKX2.2, CD99, and the chondrocytes for S100 and SOX9. The immunohistochemistry profile of the undifferentiated component is non-specific and overlaps with other round blue cell tumors. NKX3.1, though widely used as a diagnostic biomarker for prostatic adenocarcinoma, has been recently proposed by Yoshida et al. as a unique marker of mesenchymal chondrosarcoma [19]. The reciprocal translocation (11;22)(q24;q12) was first described in 1993, suggesting that, because of the presence of a similar cytogenetic abnormality, mesenchymal chondrosarcoma may belong to the wide group of small round cell tumors [20]. More recently, recurrent hairy/enhancer-of-split related with YRPW motif 1—nuclear receptor coactivator 2 (HEY1-NCOA2) and interferon regulatory factor 2 binding protein 2 gene—caudal type homeobox 1 (IRF2BP2-CDX1) fusions have been identified in mesenchymal chondrosarcomas [21,22]. There is no recurrent translocation in conventional and dedifferentiated CS, highlighting that mesenchymal chondrosarcomas have a totally different genetic background.

Similar to skeletal osteosarcomas and Ewing sarcomas, mesenchymal CS most commonly occurs in patients between 10 and 30 years old, but it may arise at any age. Males and females appear to be equally affected. It rarely arises in extraosseous soft tissues, notably meninges. Common skeletal sites include the skull and craniofacial region (especially the mandible), ribs, spine, pelvis, and lower extremities. Mesenchymal CS is an aggressive form of cancer with a poor prognosis that can spread to other areas of the body, especially the lungs, liver, lymph nodes, meninges, and other bones.

Thus, clinical presentation and outcome are totally different between conventional, dedifferentiated CS and mesenchymal chondrosarcoma, which can be considered totally different diseases.

## 3. Current Treatment Management 

### 3.1. Conventional and Dedifferentiated Chondrosarcoma

Conventional chondrosarcomas are generally considered to be resistant to conventional chemotherapy and radiotherapy. Complete surgical resection with wide margins is the cornerstone of primary management. Neoadjuvant or adjuvant chemotherapy may provide, in some studies, a survival advantage in dedifferentiated CS [17,23,24]. However, a retrospective study supplied by the European Musculo-Skeletal Oncology Society (EMSOS) with 337 dedifferentiated CS patients showed a non-significant improvement in the survival of patients receiving chemotherapy. In this study, most of the patients were under 60 years old with no metastasis and had undergone limb salvage surgery [16]. The presence of a pathological fracture was the most significant poor prognostic factor.

Most often, the chemotherapy regimen is based on a combination of doxorubicin and cisplatin or ifosfamide. In a large series of 171 patients with advanced unresectable CS from the Rizzoli Institute and Leiden University Medical Center, overall survival was observed to be 48% at 1 year and only 2% at 5 years. The use of either doxorubicin-based or non-cytotoxic chemotherapy (imatinib combined with sirolimus) in this cohort significantly prolonged the median overall survival to 20 months in contrast to 15 months for those who received no treatment [25]. In selected cases, radiotherapy may achieve encouraging results when adequate doses are delivered by conformational techniques or proton beam/carbon ion radiotherapy [26]. Clinical trial enrollment should be actively encouraged in all patients diagnosed with advanced, surgically unresectable chondrosarcoma due to the lack of consensus treatment recommendations.

### 3.2. Mesenchymal Chondrosarcoma

As mentioned previously, mesenchymal chondrosarcomas are a distinct entity with a different biological background and clinical behavior that compares to conventional and dedifferentiated chondrosarcoma. Many medical oncologists advocate a chemotherapy treatment plan such as Ewing’s sarcoma. Radiation therapy after surgery is also typically standard practice for large tumors. Other agents, including trabectedin, may be clinically valuable. A small phase II study reported a median progression-free survival (PFS) at 12.5 months (95% CI: 7.4-not reached) in the trabectedin group, and 1.0 months (95% CI: 0.3–1.0 months), in the best supportive care group in metastatic mesenchymal chondrosarcoma patients [27].

## 4. Targets and Novel Treatments Options in Chondrosarcoma

Genomic initiatives spearheaded by The Cancer Genome Atlas (TCGA) consortium have accelerated the pace of discovery for many cancers with the molecular characterization of more than 20,000 primary cancers with matched normal samples spanning 33 cancer types. Sarcomas are very heterogeneous, and no chondrosarcoma was included in the program because of a lack of readily accessible and adequate tumor tissue, as well as the rarity of this sarcoma. The most extensive cohort includes 55 conventional or dedifferentiated CS and has been performed by the MSKCC-IMPACT program [28]. As expected, the most common alterations were *IDH1/2* mutations (34%), *CDKN2A/B* deletions (34%), *TP53* mutations (33%), and *Telomerase reverse transcriptase (TERT*) alterations (20%) (Figure 2). No extensive genomic data in a large cohort included mesenchymal chondrosarcomas. Because mesenchymal chondrosarcomas harbor a recurrent translocation, the number of molecular alterations which could be efficient targets is probably lower.

A better understanding of CS genomic alterations and biology should allow new potential therapeutic options (Figure 3). Ongoing clinical trials (Table 2), as well as future potential treatments for CS management, are summarized below.

### 4.1. Angiogenesis Pathway

Angiogenesis pathways are potentially effective targets for arresting the growth and spread of chondrosarcomas. It is necessary for tumors to develop a vasculature to grow. Although originating from cartilaginous tissue, which is one of a handful of avascular tissues in the human body, chondrosarcomas have been shown to exhibit a microvascularity that has been associated with aggressive clinical behavior and a higher potential for metastasis [29,30]. The vascular endothelial growth factor (VEGF) and platelet-derived growth factor (PDGF) are essential for the neovascularization required for tumor maintenance and propagation (Figure 3). Efforts to develop antiangiogenic therapies have produced many agents, including small molecule tyrosine kinase inhibitors and fully human monoclonal antibodies which affect angiogenesis.

Pazopanib is a multitargeted tyrosine kinase inhibitor that inhibits angiogenesis pathways, namely VEGF, and has shown a significant effect on tumor vascular density, viability, and volume in mice with chondrosarcoma xenografts. Pazopanib has also been shown to provide a benefit to non-adipocytic soft tissue sarcoma patients by extending progression-free survival for several months when compared to the placebo [31].

Several studies and clinical trials in bone sarcomas are published or ongoing. The first studies focused on imatinib: a multikinase inhibitor that notably targets PDGFR. Imatinib failed to show benefits in unresectable and metastatic CS [32]. However, recent studies showed an improvement in the disease control of unresectable or metastatic conventional chondrosarcoma who received pazopanib or regorafenib: another multikinase inhibitor, which also targets VEGFRs [33,34,35,36] (Table 3). These recent results are very promising and will allow VEGFR inhibitors used in daily practice soon. Approaches based on the association of treatments are also in development, including clinical trials with chemotherapy and/or PD-L1 inhibitors associated with antiangiogenic agents (Table 2).

### 4.2. Isocitrate Dehydrogenase (IDH) Mutations

Isocitrate dehydrogenase 1 (IDH1) and Isocitrate dehydrogenase 2 (IDH2) are enzymes that catalyze the reversible oxidative decarboxylation of isocitrate to yield α-ketoglutarate (α-KG) as part of the tricarboxylic acid cycle in glucose metabolism. Indeed, IDH1 and IDH2 participate in fatty and glucose metabolism and oxidative damage. *IDH1/2* mutations are present in 50% of central CS and 60% of dedifferentiated CS [39] (Figure 3). The presence of IDH mutations in benign enchondromas and malignant CS supports the notion that IDH mutations are early events, and these cartilaginous neoplasms represent a spectrum of malignant potential. IDH mutations are also found in gliomas, acute myeloid leukemia (AML), and cholangiocarcinomas [40]. These mutations are considered driver events in these types of cancers too. *IDH1* and *IDH2* mutations are recurrent and typically involve an amino acid substitution in the active site of the enzyme in codon 132 (IDH1) and in codon 172 (IDH2). Specific inhibitors are in development. In AML, the US Food and Drug Administration (FDA) has already approved ivosidenib as the first IDH1 inhibitor for patients with relapsed or refractory AML and an *IDH1* mutation. New IDH1 inhibitors, as well as dual IDH1 and IDH2 inhibitors, are in development. These drugs are currently evaluated for chondrosarcoma patients (Table 2).

### 4.3. Immunotherapy Strategies

In 2016, Kostine M. et al. reported an immunohistochemical analysis of PD-L1 expression in a large series of conventional, mesenchymal, and dedifferentiated chondrosarcomas using tissue microarrays. PD-L1 expression was absent in conventional and mesenchymal chondrosarcomas. Forty-one percent (9 of the 22) of dedifferentiated chondrosarcomas displayed PD-L1 positivity. In this study, PD-L1 expression was exclusively found in the dedifferentiated component and its expression positively correlated with a high number of tumor-infiltrating lymphocytes (*p* = 0.014) and positive HLA class I expression (*p* = 0.024) but not with patient overall survival (*p* = 0.22) [15]. Yang X et al. reported that the positivity for the PD-L1 and PD-L2 expressions in chondrosarcoma were 68% and 42%, respectively [41] (Figure 3). A 2020 study published by Iseulys and colleagues found that tumor-associated macrophages were the predominant immune cell type in the immune environment of chondrosarcoma [42]. High levels of CD68^+^ macrophages were associated with metastatic disease at diagnosis and a poor prognosis. The authors also reported the increased expression of the colony-stimulating factor 1 receptor (CSF1R), signal regulatory protein alpha (SIRPA), B7 superfamily member-H3 (B7H3), T cell immunoglobulin mucin (TIM3), and lymphocyte activation gene-3 (LAG3). Although low levels of the tumor mutational burden (TMB) have been reported in chondrosarcoma [43], TMB seems associated with histological grade. Grades 2 and 3 and dedifferentiated chondrosarcomas show levels of somatic mutations more than two times higher than grade 1 CS [44].

To date, only a few clinical studies that focused on immune checkpoint inhibitors in sarcoma patients are available. SARC028 was the first multicenter, open-label phase 2 clinical trial evaluating pembrolizumab monotherapy in 85 patients with advanced soft tissue and bone sarcomas. Only five patients with chondrosarcoma were enrolled in this study, and one patient with dedifferentiated CS achieved a partial response [37] (Table 3).

In a non-randomized phase 1/2 clinical trial of 37 patients with advanced sarcoma, the combination of doxorubicin and pembrolizumab was well tolerated. Three out of eight patients with chondrosarcoma had tumor regression, including one conventional chondrosarcoma with a 26% decrease in size [45]. Taken together, further evaluation of immune checkpoint blockades in chondrosarcomas is warranted. Several clinical trials are currently ongoing (Table 2).

### 4.4. Cell Cycle Pathway

The cyclin-dependent kinase inhibitor 2A (CDKN2A) and cyclin-dependent kinase inhibitor 2B (CDKN2B) act as tumor suppressors by regulating the cell cycle; they encode p16 and p15Ink4b, respectively. They inhibit cyclin-dependent kinases 4 and 6 (CDK4 and CDK6) and thereby activate the retinoblastoma family of proteins, which block traversal from G1 to S-phase. CDKN2A and CDKN2B deletions are recurrent in chondrosarcomas as well as in several subtypes of cancers. Palbociclib, a CDK4/CDK6 inhibitor, was approved to be used in combination with an endocrine therapy to treat advanced-stage or metastatic, hormone-receptor-positive, HER2-negative breast cancer patients regardless of CDK4N2A/B status. There is no validated predictive biomarker for CDK4/6 inhibitor response, including CDKN2A/B deletions. The first reports of CDK4/6 used in other subtypes of cancer seem to show that Palbociclib monotherapy does not have clinical activity [46]. Several studies are ongoing, including studies with new CDK4/6 inhibitors for chondrosarcoma patients (Table 2).

### 4.5. PI3K-AKT-mTOR Pathway

mTOR is a serine/threonine tyrosine kinase that acts as a key regulatory protein in normal cell growth, development, metabolism, as well as angiogenic pathways [47]. The (PI3K)-AKT signaling network has many direct and indirect downstream effects on cellular metabolism [48]. The activation of the downstream PI3K/AKT pathway has been demonstrated in CS cells [48] (Figure 3). Ten consecutive patients with unresectable CS were offered off-label treatment with a sirolimus/cyclophosphamide combination, which was well tolerated and had modest clinical activity [49]. One (10%) objective response was observed, and six (60%) patients experienced the stabilization of the disease for at least 6 months. Three patients had progressive disease.

Temsirolimus may also potentiate the cytotoxicity of liposomal doxorubicin [50]. Lastly, neoadjuvant everolimus has been tried in CS (NCT02008019, CHONRAD). The study was suspended due to limited activity. Preclinical data indicate that the combination of mTOR with an IGF-1R blockade results in greater AKT downregulation and enhanced antiproliferative effects. More encouraging results were achieved in combination studies, and partial responses were shown in CS cases treated with a dual IGF1R/mTOR blockade [38].

### 4.6. Epigenetic Strategies

Epigenetic regulation is critical to the physiological control of development, cell fate, cell proliferation, genomic integrity, and, more specifically, transcriptional regulation. Post-transcriptional gene regulation is caused by DNA methylation and histone modification. Mutations in *IDH1/2* genes result in progressive increases in DNA and histone methylation and are observed in 50% of conventional chondrosarcomas, suggesting that epigenetic dysregulation represents a potential barrier for tumor progression and a target for therapeutic intervention [51]. Venneker S. et al. reported that high-grade tumors demonstrated an increased number of hypermethylated genes compared to low-grade tumors, suggesting that epigenetic mechanisms play an important role in CS progression [52]. A broad compound screen that targets different epigenetic key players (including histone deacetylases (HDACs), sirtuins (SIRTs), histone demethylases (HDMs), histone acetyltransferases (HATs), histone methyltransferases (HMTs), and DNA methyltransferases (DNMTs)) was performed on IDH wild type and mutant CS cell lines to explore whether these epigenetic changes could be used as a target for novel anti-cancer therapy. Interesting compound classes included HDAC and bromodomain, and extra-terminal motif (BET) protein inhibitors were identified (Figure 3). For example, it has been shown that the HDAC inhibitors romidepsin, trichostatin A, and sodium valproate affect cell proliferation in 2D in vitro models of CS.

The analysis of 92 central and 45 peripheral CS tumors also showed that although the CS were strongly positive for the H3K4me3, H3K9me3, and H3K27me3 histone modifications, neither of these modifications nor the variable levels of 5-hydroxymethylcytosine and 5-methylcytosine correlated with the *IDH1/2* mutation status [13].

### 4.7. Anti DR5

TNF-related apoptosis-inducing ligand (TRAIL) receptor (DR4 or DR5) agonists are promising agents for cancer therapy because they selectively induce apoptosis in cancer cells. DR5 is expressed in a broad spectrum of hematologic malignancies and solid tumors (Figure 3). Monoclonal antibodies targeting DR5 are under development. Among them, INBRX-109 seems to be promising for chondrosarcoma patient management. The phase 1 (NCT04553692, Table 3) results were presented at a 2020 CTOS virtual meeting: disease control was observed in 11 of the 12 patients (92%), and 8 of the 12 patients (67%) achieved objective response rates. After the success of the phase 1 trial, phase 2 should open soon.

### 4.8. Others

YAP (encoded by Yes-associated protein 1) and TAZ (encoded by WWTR1) [53,54] are transcriptional activators that regulate cell proliferation, survival, and differentiation [55]. The Hippo signaling pathway is a conserved cascade that regulates the growth and size of tissues, involving the translocation of Yorkie, the YAP/TAZ homolog, between the cytoplasm and nucleus. YAP/TAZ is also known to play a vital role in osteogenesis [56] and chondrogenesis [57]. A study showed that less than half of CS specimens harbored activated YAP/TAZ [58]. The downregulation of YAP/TAZ and LATS1 in chondrosarcoma cells treated with the BRD4 inhibitor, JQ1, led to cell cycle arrest, senescence, and apoptosis [59]. A multi-institutional, open-label, single-arm trial of dasatinib, which inhibits the nuclear localization and target gene expression of YAP and TAZ, was performed in patients with advanced sarcoma, including grade 1 or 2 CS (*n* = 33) [60]. The median PFS was 5.5 months for CS. Three patients with CS remained on treatment for more than 2 years. Six patients with CS had an objective tumor response according to the Choi criteria. Specific targeted therapies against Hippo signaling pathways should be developed in the future and could be good candidates for CS treatments.

The Hedgehog pathway (Hh) is a signaling cascade that plays a crucial role in many fundamental processes, including stem cell maintenance, cell differentiation, tissue polarity, and cell proliferation. The components of the Hh signaling pathway, involved in the signaling transfer to the Gli transcription factors, include Hedgehog ligands (Sonic Hh [SHh], Indian Hh [IHh], and Desert Hh [DHh]), Patched receptor (Ptch1, Ptch2), Smoothened receptor (Smo), Suppressor of fused homolog (Sufu), kinesin protein Kif7, protein kinase A (PKA), and cyclic adenosine monophosphate (cAMP). The misregulation of the Hh pathway has been found to lead to carcinogenesis in many cancers, including chondrosarcomas [61]. Preclinical studies showed encouraging results when using saridegib (IPI-926), an oral Hh inhibitor, in primary mice CHS xenografts [62]. Unfortunately, the clinical trials had to be stopped due to disappointing clinical data. Vismodegib is an FDA-approved Hh inhibitor for basocellular carcinoma patients. The results of clinical trials conducted with vismodegib have also been discouraging. In a single-arm phase II trial, vismodegib showed some activity in patients with grade 1 or 2 conventional chondrosarcomas. However, the primary endpoint (6 months clinical benefit rate) was met [63] (Figure 3). A selection of patients more likely to benefit from this kind of targeted therapy with surrogate markers appears to be needed for future studies.

Behind *TP53* and *IDH1/2* mutations, *COL2A1* molecular alterations (insertions, deletions, and rearrangements) have been shown to be present in 37% of chondrosarcomas [42]. *COL2A1* encodes the α-chain of type II collagen fibers: the major collagen constituent of articular cartilage. The disruption of the collagen maturation process through the production of aberrant pro-collagen α-chain is the likely result. Mutations in exostosin glycosyltransferase 1/2 (EXT1/2) genes, which participate in the differentiation of chondrocytes, have been observed in peripheral chondrosarcomas. To date, there is no targeted therapy developed against these genes.

## 5. Conclusions

Conventional, dedifferentiated, and mesenchymal CS are rare bone sarcomas with a very poor prognosis. Mesenchymal chondrosarcomas harbor recurrent translocations and are distinct pathological, biological, and clinical entities. The management is multimodal and based on polychemotherapy, radiotherapy, and surgery, as performed in Ewing sarcoma patients. Conventional chemotherapies fail to improve the survival of conventional and dedifferentiated CS patients. Conventional and dedifferentiated CS remains a primarily surgical disease at present which may respond to doxorubicin-containing chemotherapy in only rare cases. Recently, genomic programs identified recurrent alterations, especially in *IDH1/2* genes. The efficiency of IDH1 inhibitors in other malignancies suggests that these inhibitors will be part of IDH-mutated CS management soon. Other treatment approaches based on a better understanding of CS biology are emerging, such as PIK3-AKT-mTOR inhibitors, cell cycle inhibitors, and epigenetic or immune modulators.

This review highlights how a comprehensive and multimodal understanding of CS is crucial for patient management and improvement. We showed that CS patient care is based on radiologic, pathological, and clinical expertise. Recently, the understanding of the genomic background and biology of CS has opened new therapeutic options and will probably change the CS standard of care in the future.

## Figures and Tables

**Figure 1 ijms-24-01361-f001:**
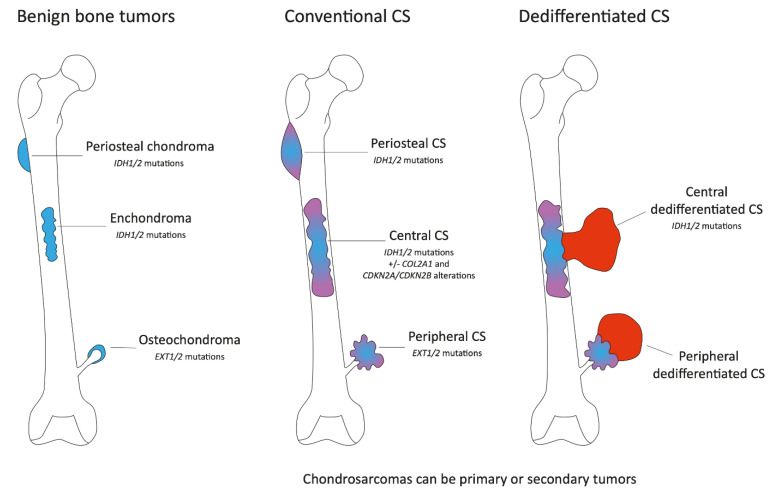
Schematic representation of conventional cartilaginous tumors of bone. The figure represents the benign bone lesions, osteochondromas (sessile osteochondroma and pedunculated osteochondroma), chondromas (enchondroma and periosteal chondroma), their location, and the frequency of somatic tumors present in these lesions (**left** part). These lesions may progress to malignant conventional chondrosarcomas (**middle** part). To note, conventional chondrosarcomas may be primary or secondary tumors. In some cases, central and peripheral conventional chondrosarcomas may progress to dedifferentiated chondrosarcomas (**right** part).

**Figure 2 ijms-24-01361-f002:**
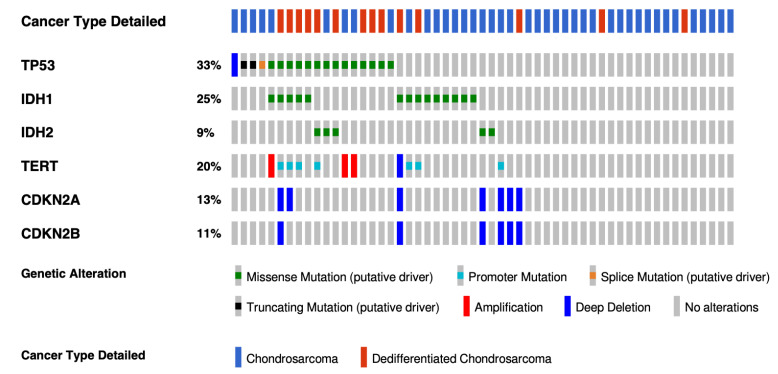
Oncoprint with the most frequently altered genes in conventional and dedifferentiated chondrosarcomas from MSKCC-IMPACT cohort. The oncoprint represents the most common alterations in 44 conventional and 14 dedifferentiated chondrosarcomas from the MSKCC impact cohort (taken from cbioportal website, https://www.cbioportal.org (accessed on 17 February 2022)).

**Figure 3 ijms-24-01361-f003:**
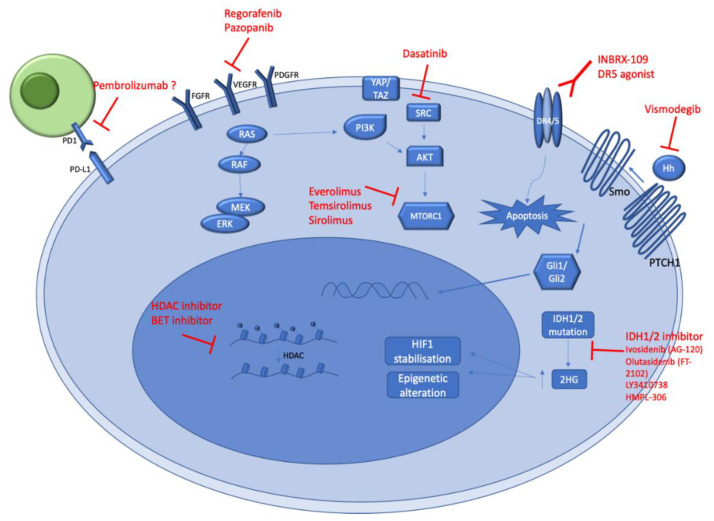
Schematic illustration of key signaling pathways underpinning chondrosarcoma genesis. The figure represents the main biological pathways (in blue) activated in conventional and dedifferentiated chondrosarcomas. The potential therapeutic agents which could target these pathways are represented in red. All the mentioned pharmacological agents in this figure are under evaluation in ongoing clinical trials.

**Table 1 ijms-24-01361-t001:** Chondrosarcoma histological subtypes. We excluded clear cell chondrosarcomas (CS) which represent around 2.5% of all CS.

Type (WHO 2020)		Biologic Marker	Incidence	O’Neal & Ackerman Grade	Prognosis
Conventional CS	Central CS	*IDH1/2* mutation	72% (mainly in flat bones and long bones)	1 (central ACT) 2 or 3	No metastasis Potential for metastasize
	Peripheral CS	No IDH1/2 mutation	13%	1 (secondary peripheral ACT) 2 or 3	No metastasis Potential for metastasize
	Periosteal CS	*IDH1/2* mutation	1.5%	Not applicable	Good
Dedifferentiated CS		*IDH1/2* mutation (except in dedifferentiated peripheral CS)	10%	Not applicable	High grade Poor prognosis
Mesenchymal CS		Recurrent fusion transcript (HEY1-NCOA2, IRF2BP2-CDX1)	1%	Not applicable	High grade Poor prognosis

CS: chondrosarcoma; ACT: atypical cartilaginous tumors; *IDH1/2:* Isocitrate dehydrogenase 1/2; *HEY1-NCOA2*: hairy/enhancer-of-split related with YRPW motif 1nuclear receptor coactivator 2; *IRF2BP2—CDX1*: interferon regulatory factor 2 binding protein 2 gene—caudal type homeobox 1.

**Table 2 ijms-24-01361-t002:** Ongoing clinical trials for chondrosarcomas patients.

Clinical Trial	NCT Number	Conditions	Drugs	Drug Classes
Trial of Sunitinib and/or Nivolumab Plus Chemotherapy in Advanced Soft Tissue and Bone Sarcomas	NCT03277924	All sarcoma subtypes	Sunitinib Nivolumab Epirubicin Ifosfamide Doxorubicin Dacarbazine Cisplatin Methotrexate	Antiangiogenic PD-L1 inhibitor Chemotherapy
Study of LY3410738 Administered to Patients with Advanced Solid Tumors with IDH1 or IDH2 Mutations	NCT04521686	Basket trial with *IDH1/2* mutations	LY3410738	IDH1 and IDH2 inhibitor
A Study of HMPL-306 in Advanced Solid Tumors with IDH Mutations	NCT04762602	Basket trial with *IDH1/2* mutations	HMPL-306	IDH1 and IDH2 inhibitor
AG-120 in People with IDH1 Mutant Chondrosarcoma	NCT04278781	Chondrosarcoma with *IDH1* Gene Mutation	AG-120	IDH1 inhibitor
A Study of FT 2102 in Participants with Advanced Solid Tumors and Gliomas with an IDH1 Mutation	NCT03684811	Basket trial with *IDH1* mutation	FT-2102 Azacitidine Nivolumab Gemcitabine and Cisplatin	IDH1 inhibitor DNA methyltransferase inhibitor PD-L1 inhibitor Chemotherapy
Safety, Tolerability, and Pharmacokinetics of an Anti-PD-1 Monoclonal Antibody in Subjects with Advanced Malignancies	NCT03474640	All tumors (Phase 1)	Toripalimab	PD-1 inhibitor
A Phase II of Nivolumab Plus Ipilimumab in Non-resectable Sarcoma and Endometrial Carcinoma	NCT02982486	Soft Tissue Sarcoma Bone Sarcoma Chondrosarcoma Gastrointestinal Stromal Sarcoma Ewing’s Tumor Metastatic Ewing’s Tumor Recurrent Osteosarcoma Desmoplastic Small Round Cell Tumor	Ipilimumab Nivolumab	CTLA-4 inhibitor PD-L1 inhibitor
IACS-6274 with or without Pembrolizumab for the Treatment of Advanced Solid Tumors	NCT05039801	Basket trial	IPN60090 Pembrolizumab	Glutaminase Inhibitor PD-1 inhibitor
LN-145 or LN-145-S1 in Treating Patients with Relapsed or Refractory Ovarian Cancer, Anaplastic Thyroid Cancer, Osteosarcoma, or Other Bone and Soft Tissue Sarcomas	NCT03449108	Basket trial	Aldesleukin Autologous Tumor Infiltrating Lymphocytes LN-145 Autologous Tumor Infiltrating Lymphocytes LN-145-S1 Cyclophosphamide Fludarabine Ipilimumab Nivolumab	Recombinant IL-2 Cell therapy Chemotherapy CTLA-4 inhibitor PD-L1 inhibitor
Autologous Dendritic Cell Vaccine in Patients with Soft Tissue Sarcoma	NCT01883518	All sarcoma subtypes	Autologous dendritic cell vaccine	Cell therapy
Study of INBRX-109 in Conventional Chondrosarcoma	NCT04950075	CS	INBRX-109	Tetravalent DR5 agonistic antibody
Phase 1 Study of INBRX-109 in Subjects with Locally Advanced or Metastatic Solid Tumors Including Sarcomas	NCT03715933	Basket trial	INBRX-109 Carboplatin Cisplatin Pemetrexed 5-fluorouracil Irinotecan Temozolomide	Antibody targeting Death Receptor 5 (DR5) Chemotherapy
Phase I Study of IGM-8444 as a Single Agent and in Combination with Subjects with Relapsed and/or Refractory Solid Cancers	NCT04553692	Basket trial	IGM-8444 FOLFIRI Bevacizumab (and approved biosimilars) Birinapant Venetoclax	Antibody targeting Death Receptor 5 (DR5) Chemotherapy Antiangiogenic Second mitochondrial-derived activator of caspases (SMAC) and inhibitor of IAP (Inhibitor of Apoptosis Protein) BCL2 inhibitor
Abemaciclib for Bone and Soft Tissue Sarcoma with Cyclin- Dependent Kinase (CDK) Pathway Alteration	NCT04040205	Chondrosarcoma Osteosarcoma Soft Tissue Sarcoma	Abemaciclib	CDK4/6 inhibitor
Sirolimus and Cyclophosphamide in Metastatic or Unresectable Myxoid Liposarcoma and Chondrosarcoma	NCT02821507	CS Myxoid Liposarcoma Mesenchymal Chondrosarcoma Dedifferentiated Chondrosarcoma	Sirolimus and cyclophosphamide	mTOR inhibitor and chemotherapy
TQB3525 for Advanced Bone Sarcomas with PI3KA Mutations or PTEN Loss	NCT04690725	All tumors (Phase 1)	TQB3525	PI3Ka inhibitor
Multicohort Trial of Trabectedin and Low-dose Radiation Therapy in Advanced/Metastatic Sarcomas	NCT05131386	Soft Tissue Sarcoma Bone Tumors Small Round-cell Sarcomas	Trabectedin	Chemotherapy
Testing the Combination of Belinostat and SGI-110 (Guadecitabine) or ASTX727 for the Treatment of Unresectable and Metastatic Conventional Chondrosarcoma	NCT04340843	Locally Advanced Unresectable Primary Central Chondrosarcoma Metastatic Primary Central Chondrosarcoma Unresectable Primary Central Chondrosarcoma	Belinostat Decitabine and Cedazuridine Guadecitabine	HDAC inhibitor Antimetabolites Anitmetabolite
Vismodegib in Treating Patients with Advanced Chondrosarcomas	NCT01267955	All chondrosarcoma subtypes	Vismodegib	Hedgehog pathway inhibitor
Itacitinib in Treating Patients with Refractory Metastatic/Advanced Sarcomas	NCT03670069	All sarcoma subtypes	Itacitinib	JAK-1 inhibitor

**Table 3 ijms-24-01361-t003:** Published clinical trials.

Trial	Trial Phase	Intervention	Type of Tumor	Years of Inclusion	Geographic Distribution	Median Age	N	Previous Lines of Treatment	CR (N)	PR (N)	SD (N)	PD (N)	RR (%)	DCR (%)	Median PFS (Months)	Median OS (Months)
**Grignani G. et al.** **Cancer 2011** [32]	II	**Imatinib**	CS	**2007–2009**	Multicenter Italy	61	26	At least 1: 100%	0	0	8	18	0	30	3	11
**Chow W. et al.** **Cancer 2020** [33]	II (1 arm)	**Pazopanib**	CS	2011–2015	Multicenter USA/UK	58	47	At least 1: 32%	0	1	30	11	2	65	7.9	17.6
**Tap W. D. et al.** **J Clin Oncol 2020** [34]	I	**Ivosidenib**	CS	March 2014	Multicenter	55	21	>1: 50% <1: 50%	0	1	11	6	4	57	5.6	NA
**Liu Z. L. et al.****Cancer Med 2021** [35]	Restrospective	**Anlotinib**	Bone sarcoma	2018–2020	China	24	9	1: 37 2: 31 3: 12	0	0	7	2	0	77	4.2	NA
**Duffaud F. et al.****Eur J Cancer 2021** [36]	II	**Regorafenib**	CS	2014–2019	Multicenter France	64	24	1: 17 2: 7	0	2	11	10	8	54	5	11.7
**Tawbi H.A. et al.** **Lancet Oncol 2017** [37]	II (1 arm)	**Pembrolizumab**	Soft Tissue/bone sarcoma	2015–2016	Multicenter	33 (bone sarcoma)	5 (CS)	1: 19% 2: 38% 3: 43%	0	1	1	3	20	32	8	12
**Schwartz G.K. et al.** **Lancet Oncol 2013** [38]	II	**Cixutumumab + temsirolimus**	Bone sarcoma	**2009–2012**	Multicenter USA	47	17	<2: 50% >2: 50%	0	1	NA	NA	6	NA	5.2	13.6

CS: chondrosarcoma; CR: complete response; PR: partial response; SD: stable disease; PD: progressive disease; RR: response rate; DCR: disease control rate; PFS: progression free survival; OS: overall survival.

## Data Availability

Data available within the article.

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
