# Peer review of "Biology and Management of High-Grade Chondrosarcoma: An Update on Targets and Treatment Options"

_ijms, 2023, doi:10.3390/ijms24021361_

Round 1

Reviewer 1 Report

The manuscript deals with a topic of great impact for Oncology and Orthopaedic surgery. It is complete and very well written so it is worthy of publication.

Author Response

We thank the reviewer for their positive comments.

Reviewer 2 Report

This is a comprehensive and well-organised review, the authors should be congratulated

Author Response

(The authors gave the same response as above.)

Reviewer 3 Report

Dear Editor,

I reviewed the manuscript by Telmisani et al., entitled "Biology and Management of high-grade Chondrosarcoma: an 2 update on Targets and Treatment options "

It is a very interesting work tha deals with substanial information about the biology of chondrosarcoma and the mangment of this disease. The manuscript is well written and provides important information for the reader of the journal. Accordingly. This manuscript can be published in present form after minor spell check of the text.

Many thanks.

Author Response

We thank the reviewer for their positive comments. We carefully read the manuscript to improve the English language. 

Reviewer 4 Report

The paper is on histopathology, clinical presentation, molecular pathways, and treatments of chondrosarcomas. It is a nearly comprehensive clinical review but suffers from the poor writing style and general information. the paper is actable considering revision:

1. write the gene's complete name

2. write a paragraph explaining the necessity or novelty of your review

3. delete  treatment and novel agents from keywords, unrelated

4. too short paragraphs in the introductions . rewrite these paragraphs

5. rewrite the introduction. it does not have consistency

6. page 5, line 192

Current management of chondrosarcoma to  Current treatment management...

7. remove the links in table 2 and add citations to the studies instead

8. figure 2:  put a citation instead of https://www.cbioportal.org

9. add citations in table 3

10. rewrite conclusion

Author Response

Ref. No.: IJMS-2072338

Reviewer #4: The paper is on histopathology, clinical presentation, molecular pathways, and treatments of chondrosarcomas. It is a nearly comprehensive clinical review but suffers from the poor writing style and general information. the paper is actable considering revision:

Write the gene's complete name
R: We agree with the reviewer and added the gene’s complete name under table 1; Page 3 lines 102, 121, 124; Page 5 lines 157, 178, 179; Page 6 line 233.
2. Write a paragraph explaining the necessity or novelty of your review
R: We agree with the reviewer and included this paragraph page 2 line 73 in the Introduction section:
Understanding the different histological subtypes of CS is helpful in predicting biologic behavior. In this review, a summary is given about CS clinical behavior, pathological characteristics and current treatment modalities. In the following, we will also discuss the molecular pathways in high grade CS including conventional CS, dedifferentiated CS and mesenchymal CS, including schematic illustration of key signaling pathways underpinning CS genesis. Furthermore, published and ongoing clinical trials for CS patients will be presented.

3. Delete  treatment and novel agents from keywords, unrelated
R: Thank you for your comment. The last part of the article is focused on new treatments approaches according to the molecular pathways and ongoing clinical trials. Moreover, in the second part of the manuscript, we summarize the current clinical management according to CS subtypes. In conclusion, to be more accurate and answer to your comment, we agree to change the key words. We removed “treatments” and “novel agents” and put “clinical management”, “clinical trials”, and “molecular biology” keywords instead. 
4. Too short paragraphs in the introductions. rewrite these paragraphs
R: We agree with the reviewer and rewrote th paragraphs of  the introduction in order to be more concise, consistent and to highlight the necessity of our review (see track changes in the revised manuscript attached). 

5. Rewrite the introduction. it does not have consistency
R: We agree with the reviewer and rewrote the introduction as mentioned above (see revised manuscript attached).
6. page 5, line 192 : Current management of chondrosarcoma to Current treatment management
Thank you for the suggestion. We changed the section title. 
7. Remove the links in table 2 and add citations to the studies instead
R: We agree with the reviewer and removed the links in table 2; NCT number has been specified in table 2 to find ongoing clinical trials. 
8. figure 2:  put a citation instead of https://www.cbioportal.org
R: We added the reference of the MSKCC cohort we used to provide the molecular analysis. We think this is important to keep the website adress but as requested we also add the citations related with the cbioportal website tool.  
9. Add citations in table 3
R: We agree with the reviewer and added citations in table 3. 
10. Rewrite conclusion
R: We agree with the reviewer and rewrote the conclusion (see revised manuscript attached).  

Awaiting your reply, please accept my best regards,

Dr P Boudou-Rouquette

Round 2

Reviewer 4 Report

The manuscript is acceptable regarding the authors' revision.